# Pulmonary Telerehabilitation in COPD Patients: A Systematic Review to Analyse Patients’ Adherence

**DOI:** 10.3390/healthcare13151818

**Published:** 2025-07-25

**Authors:** Pauline Aubrat, Eloïse Albert, Melvin Perreaux, Veronica Rossi, Raphael Martins de Abreu, Camilo Corbellini

**Affiliations:** Department of Health, LUNEX University of Applied Sciences, 4671 Differdange, Luxembourg; aubrat.pauline@free.fr (P.A.); eloisealbert01@gmail.com (E.A.); melvin.perreaux2@gmail.com (M.P.); veronica.rossi@policlinico.mi.it (V.R.); rmartinsdeabreu@lunex.lu (R.M.d.A.)

**Keywords:** telerehabilitation, pulmonary rehabilitation, quality of life

## Abstract

**Introduction**: Limited access to pulmonary rehabilitation (PR) has contributed to the rise of telerehabilitation (TPR) for COPD patients. Positive comparable effects are observed in exercise tolerance, quality of life (QoL), and dyspnoea with TPR. However, patient adherence to TPR is an outcome that has not been sufficiently analysed. **Objective**: To analyse adherence, satisfaction, and quality-of-life improvements in COPD patients following the TPR program to determine whether telerehabilitation is comparable to conventional therapy or usual care. **Methods**: A systematic search was conducted using four electronic databases, retrieving 392 articles. Two independent researchers selected and evaluated these articles based on predefined eligibility criteria. A third researcher was consulted in the event of disagreements. **Results**: Primary outcomes: Adherence to PR and/or usual care showed a minimum reported value of 62% and a maximum reported value of 91%, while TPR adherence had the lowest reported value of 21% and the highest reported value of 93.5%. Five articles compared TPR to PR and/or usual care, showing that TPR adherence is higher or similar to other interventions, whereas only one article found lower TPR adherence compared to PR. Secondary outcomes: A higher number of dropouts were reported for PR and usual care compared to TPR. Three publications analysed satisfaction and demonstrated that patients are satisfied across groups. Tertiary outcomes: Comparable improvements in QoL were found for TPR and PR, both being superior to usual care. **Conclusions**: This systematic review reveals heterogeneity in classifying adherence for pulmonary rehabilitation and telerehabilitation. Adherence classification may be standardised in future studies for consistent analysis.

## 1. Introduction

Chronic Obstructive Pulmonary Disease (COPD) is the third leading cause of death, imposing a considerable financial burden on society [1]; it caused 3.23 million deaths in 2019 [2] and is projected to affect 600 million people by 2050 [3]. COPD, a chronic lung illness, is defined as an irreversible, persistent airflow restriction and gas exchange impairment, commonly caused by exposure to noxious gasses [4]. Major indicators of poor health and premature death caused by COPD include quality-of-life (QoL) deterioration, dyspnoea, chronic cough, sputum production, and decreased exercise tolerance [5].

COPD management strategies include smoking cessation, lifestyle modifications, and exercising [6]. The American Thoracic Society (ATS) recommends pulmonary rehabilitation (PR) as the most effective treatment for COPD [4]. The cornerstones of PR are tailored therapies, including upper and lower extremity resistance exercise, endurance training such as cycling or treadmill activities, and patient education [7]. For optimal results, PR is administered in the hospital two to three times a week for six to eight weeks, with a maximum duration of twelve weeks [5,8]. The benefits of PR include enhancing QoL by increasing exercise capacity, improving dyspnoea, enhancing psychological well-being, and reducing the need for medical attention and hospital stays [7].

Less than 15% of eligible patients worldwide benefit from PR, highlighting a significant issue in ensuring adequate access to these programs [9]. The proportion of patients not receiving rehabilitation is attributed to limited access to centres in rural areas, lack of knowledge among health professionals, and heterogeneous programs that do not always meet patients’ needs. According to ATS, telerehabilitation (TPR) is a therapeutic approach to consider, allowing for greater accessibility [10].

Since the SARS-CoV-2 pandemic, PR has been halted due to social distancing rules, while TPR has emerged as a treatment option for COPD patients [11,12]. Regardless of a patient’s proximity to rehabilitation services, TPR utilises electronic information and communication technologies to deliver rehabilitation at home [13]. The basis of TPR includes tailored exercises such as resistance training with materials like dumbbells or elastic bands, endurance training on a treadmill or cycle ergometer, and self-management lifestyle education [8,13,14]. Patients can follow video-facilitated exercises [15] or engage in real-time videoconferencing with physiotherapists [16] to perform exercises either in groups [17] or individually [14]. Positive comparable effects are observed in exercise tolerance, QoL, and dyspnoea with both TPR and conventional centre-based PR [18]. Patient adherence to telerehabilitation remains an outcome that has not been sufficiently studied [13].

The WHO defined adherence as “the extent to which the persons’ behavior corresponds with agreed recommendations from a healthcare provider” [19,20]. Outcomes of interest include adherence rates, questionnaires to gauge patient satisfaction, and standardised quality of life (QoL) scales as: the COPD Assessment Test (CAT), St. George’s Respiratory Questionnaire (SGRQ), Euro-QoL-5 dimension (EQ-5D), and the modified Medical Research Council (mMRC). 

The adherence rate can be calculated by comparing attended sessions to the overall expected number of sessions or by tracking exercise duration against the recommended average weekly training time. However, heterogeneity in adherence measurement remains due to the lack of a standard method of quantification. Various methods exist to gauge patient satisfaction; one widely used option is the Client Satisfaction Questionnaire. Patients’ QoL will improve as a result of strong adherence [21]. The term “quality of life” is a difficult concept to define; it can be understood as an individual’s assessment of their level of satisfaction with life, encompassing their ability to pursue personal goals and well-being while considering both positive and negative aspects of their life at a given moment in time [22]. 

Therefore, in this systematic review, the primary aim is to quantify adherence, the secondary aim is to analyse patient satisfaction and dropout, and the tertiary aim is to examine QoL among COPD patients who underwent TPR in order to assess whether telerehabilitation can be similar to conventional therapy and usual care.

## 2. Materials and Methods

This systematic review was registered on PROSPERO (CRD42024572712) and conducted according to PRISMA guidelines (https://www.prisma-statement.org/, consulted on 29 March 2024).

### 2.1. Search Strategy and Study Selection

The electronic search began on 1 April 2024, and ended in June 2024, searching in PubMed, Cochrane, CINAHL, and Bibliothèque nationale du Luxembourg (BnL), using the following search terms combined with Boolean operators: “Pulmonary Disease, Chronic Obstructive” [Mesh] OR COPD” AND “Telerehabilitation OR tele-rehabilitation OR telehealth” AND “Pulmonary rehabilitation” AND “Exercise tolerance OR Patient satisfaction OR quality of life OR treatment adherence AND compliance”. This strategy helped narrow the focus of the search, yielding a total of 392 articles. The search was limited to articles published after 2014 to ensure recent and clinically relevant data, particularly in light of the significant increase in telerehabilitation methods following the COVID-19 pandemic. This approach initially retrieved all relevant articles. The identification of Randomised Controlled Trials (RCTs) was conducted manually by screening the retrieved articles, as only RCTs were considered for inclusion. Finally, 30 articles were retrieved from PubMed, 59 from Cochrane, 7 from BnL, and 1 from CINAHL. After removing duplicates and excluding titles or abstracts unrelated to the subject, as well as articles without full-text availability, the remaining articles were examined in full text, and only those meeting the inclusion criteria were selected for this systematic review. Two independent researchers (A.E. and A.P.) selected the studies and evaluated each article by first reviewing the abstract and title, after which the potentially eligible articles were analysed in full text for inclusion. A third researcher (P.M.) was consulted in case of disagreements throughout this process.

### 2.2. Eligibility Criteria

Inclusion criteria were determined based on Population, Intervention, Comparison, and Outcomes (PICO). Following spirometry tests, the American Thoracic Society (ATS)/European Respiratory Society (ERS) or Global initiative for Obstructive Lung Disease (GOLD) criteria [23] are commonly used to classify Chronic Obstructive Pulmonary Disease (COPD) severity. This systematic review included participants over 18 years old with a COPD diagnosis regardless of severity. The intervention of interest was telerehabilitation versus conventional pulmonary rehabilitation (PR) and/or usual care. Primary outcomes of interest included patient adherence and compliance. Secondary outcomes encompassed patient satisfaction and dropout rates, while tertiary outcomes focused on quality of life (QoL). Articles had to be written in English, French, or Spanish and be available in full text. Exclusion criteria included all trials without randomisation or those that failed to fulfil the PICO criteria, such as mixed samples not exclusively comprising COPD participants.

### 2.3. Data Extraction

Data from the enrolled studies was extracted using study-specific forms with the following domains: authors, publication date, country, sample size, gender, age, COPD severity, baseline QoL scale score, rehabilitation program duration, and sessions attended. Additionally, detailed intervention characteristics were collected, such as the frequency and duration of TPR and PR program sessions, materials used, and delivery method. Outcome data were categorised into primary, secondary, and tertiary measures and extracted accordingly for further analysis.

### 2.4. Quality Assessment

The methodological quality and scientific rigour of the included studies were assessed using the PEDro scale [24] for methodological robustness, supplemented by the CASP checklist for RCTs [25] to evaluate clinical relevance and applicability. This dual approach provided a more holistic quality appraisal. The PEDro scale comprises 11 questions based on eligibility criteria, randomisation and concealed allocation, baseline comparability, blindness of subjects, therapists and assessors, follow-up, intention-to-treat analysis, between-group comparability, and measures of variability. Each aspect is scored 0 or 1, corresponding, respectively, to yes or no, except for eligibility criteria, which receive no score. Studies receive a maximum score of 10, with scores of 9–10, 6–8, 4–5, and 0–3 indicating excellent, good, fair, and poor quality, respectively. The CASP checklist also consists of 11 questions, evaluating the clarity of research questions, randomisation method, loss to follow-up, intention-to-treat analysis, blindness of participants, therapists and assessors, baseline comparability, group comparability, group equality, results analysis, and results generalisability. The results of the quality assessment are presented in the corresponding tables.

### 2.5. Certainty of Evidence

The Grading of Recommendations, Assessment, Development, and Evaluations (GRADE) [26] approach was used to evaluate the certainty of study evidence, with the results summarised in a dedicated table. Factors such as the risk of bias, imprecision, inconsistencies, indirectness, and publication bias reduce the certainty of the evidence. GRADE consists of four levels of evidence: high, moderate, low, and very low. High certainty indicates that no further research can change our confidence in the effect estimate, and all five domains are satisfied. Moderate quality means that four out of five domains are satisfied; additional research may affect our confidence and potentially change the effect estimate. Low quality indicates that three out of five domains are satisfied; further research is very likely to impact our confidence and modify the effect estimate. Very low quality refers to two out of five domains being satisfied, where any effect estimate remains highly uncertain.

## 3. Results

Three hundred ninety-two manuscripts were retrieved from PubMed, Cochrane, BnL, and CINAHL. Only Randomised Controlled Trials (RCTs) and publications from the last ten years were selected, yielding 97 articles. This process excluded 27 duplicate articles, along with 30 articles that did not meet the eligibility criteria after review of the title and abstract, and 23 articles with no full text available, leaving 17 articles to be screened for full-text availability. Applying the eligibility criteria to the remaining 17 articles, 8 articles that did not correspond to Population, Intervention, Comparison, and Outcomes (PICO) were excluded from PubMed and 1 from Cochrane. The main reasons for exclusion following the study analysis were the failure to assess adherence as a quantitative outcome and the absence of telerehabilitation as an intervention. Finally, eight articles were included in this systematic review (Figure 1).

### 3.1. Studies Characteristics

The main characteristics of each study are presented in Table 1. This systematic review includes papers published between 2014 and 2023. All studies except for Tabak et al. (2014) [28] and Tsai et al. (2017) [29] acquired signed consent forms from subjects. All of them received ethical committee approval. The total sample size consisted of 664 subjects, with the number of participants in intervention and control groups ranging from 12 to 67. The mean age of participants ranged from 62 to 75, encompassing both men and women. Chronic Obstructive Pulmonary Disease (COPD) severity was classified according to Global Initiative for Chronic Obstructive Lung Disease (GOLD) criteria, and the mean predicted Forced Expiratory Volume in one second (FEV1) percentage [23]. In the selected studies, COPD severity ranged from mild to very severe airway obstruction based on GOLD criteria, with the highest mean predicted FEV1 being 68 and the lowest 32.6, indicating that the average COPD severity ranged from moderate to severe. The duration of the studies varied from 42 [30] to 365 days [31,32]. Four of them compared telerehabilitation (TPR) to pulmonary rehabilitation (PR) [30,33,34,35], three compared TPR to usual care [28,29,31], and one compared TPR to both PR and usual care [32]. Adherence was measured using the mean number of sessions attended per week [30], the number of exercise sessions/appointments attended based on the scheduled ones [31], participation in all scheduled sessions [33], the number of sessions performed relative to the expected sessions [29,32,34], the number of exercises prescribed versus the number performed [28], and the minutes of exercise achieved compared to the expected minutes [35].

Several quality-of-life (QoL) scales were employed: seven studies incorporated the modified Medical Research Council (mMRC)/Medical Research Council (MRC) [28,29,30,31,32,33,34], six applied the COPD Assessment Test (CAT) [29,30,31,32,33,34], three used the St. George’s Respiratory Questionnaire (SGRQ) [30,32,35], two utilised the Chronic Respiratory Disease Questionnaire (CRQ) [29,31], one included SF-36 [31], and two used both Euro-QoL 5 and the Clinical COPD Questionnaire (CCQ) [28,33].

### 3.2. Intervention Characteristics

Each TPR program is based on exercises comparable to conventional PR, including aerobic exercises and/or resistance training, generally complemented by patient education. Most studies in the TPR group incorporated patient education [28,30,32,33,34,35], with the exception of two [29,31]. Five studies presented TPR exercises using pre-recorded videos [28,30,31,32,35], while three studies involved a physiotherapist in real time [29,33,34]. TPR was delivered through smartphones [31,35], tablets [32,34,35], and laptop computers [29]. Although TPR deliveries were conducted online, three RCTS did not provide sufficient details about the electronic devices used [28,30,33]. One study failed to specify the TPR materials utilised [30]. The duration of TPR exercises varied across studies, ranging from 10 min to an hour and a half, with one study not reporting the duration [28]. The frequency of exercise varied from two to a maximum of five times per week. Four studies included usual care programs focused on self-management [28,29,31,32]. In the selected articles, PR exercises were often performed twice a week, and five of them included educational sessions [30,32,33,34,35]. Training durations were comparable to those of the TPR program except for Cerdán-De-Las-Heras et al. (2021) [35], where participants in the TPR group received a 20 min training session compared to 60 min in the PR group. All details about the interventions are presented in Table 2.

### 3.3. Results of Primary Outcome

Adherence to PR was reported with the lowest value of 62% across articles, whereas adherence to TPR showed more variable findings, with reported values from the lowest at 21% to the highest at 93.5%. One article [35] did not provide a comparison between TPR and PR, and three articles [28,29,32] did not compare TPR to usual care regarding exercise adherence. However, two of these articles [29,35] reported high TPR adherence rates, above 80% (82% to 90%, mean 91.7%, respectively), while one RCT [28] found a lower TPR exercise adherence rate of 21%. According to Polo et al. (2023) [34], TPR is similar in the context of patient adherence to sessions when compared to PR (59.21% vs. PR 62.95%). Only one article [30] showed adherence to TPR exercise sessions lower than PR (62% vs. 72%), with attendance to PR being more stable over time, showing no drop in participation (week 1: 1.6 sessions/week 6: 1.4); however, a decrease in participation was observed for TPR (week 1: 3.6 sessions/week 6: 2.5).

In contrast, Cerdán-De-Las-Heras et al. (2021) [35] found that TPR adherence to exercise programs increased over time (8 weeks: 82%/3 months: 90%) and that those patients exercised for longer periods (8 weeks: 394 min/6 months: 614 min), with adherence reported as moderate to high. Tsai et al. (2017) [29] demonstrated high TPR adherence, with a mean of 22 sessions performed out of 24 prescribed. In one study [31], adherence to follow-up appointments was 92.4% in the TPR group compared to 84.4% in the usual care group, with TPR session compliance at 60%. Hansen et al. (2020) [33] reported that the adherence rate for TPR was higher than for PR (73% vs. 62%), although the between-group differences were not significant, with 73% of patients attending 70% or more of the scheduled sessions (49/67 sessions). In contrast, for PR, 62% of patients attended 70% or more of the scheduled sessions, resulting in fewer sessions performed (42/67 sessions) [33]. The highest TPR adherence rate reported was 93.5% (described as excellent), corresponding to the study’s overall TPR program adherence, while adherence to TPR exercise rate was very good (91.7%) compared to 91% for PR adherence [32]. All information related to the studies’ outcomes is presented in Table 3.

### 3.4. Results of Secondary Outcomes

Hansen et al. (2020) [33] reported the highest number of dropouts in the PR group, with 24 participants leaving compared to 10 in TPR. The most frequent reason cited was low motivation (nine for PR vs. four for TPR). Three other studies [28,31,35] indicated more dropouts in PR/usual care; in one of them [35], 19 dropouts occurred in PR compared to 12, although the reasons mentioned were broad and did not specify which groups were involved. In the second study [28], usual care had 10 dropouts compared to 2 in TPR, with several reasons given (too much effort, exacerbations, rehabilitation program). The last study [31] reported eight dropouts in usual care versus five in TPR. In contrast, one study [30] reported higher dropout rates for TPR compared to PR, with 18 withdrawing (18/64) versus 5 (5/26), mainly due to voluntary withdrawals. One study [34] found equal dropout rates for TPR and PR, while Vasilopoulou et al. (2017) [32] reported no dropouts during the 12-month maintenance program. The most common reasons for dropping out in the selected articles are similar for TPR, PR, and usual care.

Three studies provided information about patient satisfaction [28,34,35]. One article [35] reported that TPR scored 4.27 on a five-point Likert scale, indicating that patients are satisfied with TPR. In the second study [28], good satisfaction was observed for TPR and usual care based on the Client-satisfaction-8 questionnaire (3 months: 26.3 TPR/29.9 usual care based on a total score of 32). In the third study [34], both groups showed high satisfaction, with no significant difference in patient satisfaction between TPR and PR based on a satisfaction survey.

### 3.5. Results of Tertiary Outcomes

The two non-inferiority studies, Bourne et al. [30] and Cerdán-De-Las-Heras et al. [35], demonstrated no difference in improvements in QoL between TPR and PR, resulting in comparable effects on the mMRC [30] and SGRQ scales [30,35]. Based on other quality-of-life scales (mMRC, CAT, Euro-QoL-5), two studies [33,34] reported no clinically meaningful difference in QoL improvements, with one study referring to the concept of the minimal clinically important difference (MCID) [33]. Improvement in QoL (higher Euro-QoL scores) for TPR and deterioration in QoL for usual care were observed [28]. In a 12-week trial [31], significant clinical improvements were noted for SF-36 MCS and CRQ-E, with significantly different time profiles between groups, particularly at 3 months for CRQ-D and CRQ-M, in favour of TPR. In another 12-week study [32], both PR and TPR maintained initial clinically meaningful improvements in SGRQ, CAT, and mMRC over usual care. Tsai et al. [29] reported a significant improvement in the CRQ total score within the TPR group and a trend toward a significant improvement of eight points for TPR over usual care. However, between-group differences for CAT were not statistically significant [29].

### 3.6. Results of Quality Assessment

In this systematic review, eight articles were included in the final analysis. All the quality assessment details are presented in Table 4. The average score of all included studies is 6.4/10, indicating good quality evidence, with PEDro scores [24] ranging from 4 to 8/10. The studies by Cerdán-De-Las-Heras et al. [35] and Vasilopoulou et al. [32] had fair quality evidence (4/10 and 5/10, respectively), while six studies demonstrated good quality evidence (from 6/10 to 8/10), reflecting good methodological quality in the majority of studies [28,29,30,31,33,34]. All included articles underwent random allocation; however, five articles used concealed allocation [28,29,30,31,33]. Four studies had blind assessors to measure at least one outcome [29,30,33,34]. All studies exhibited similar baseline characteristics and reported the results of between-group statistical comparisons for at least one key outcome. The articles did not blind any participants or therapists, as continuous interactions are required in telerehabilitation.

Considering the CASP [25] criteria, all studies defined a clear research question. All participants enrolled in the studies were accounted for in the conclusions, except for one where the number of participants at baseline differed from the number included in the final analysis, indicating that not all participants were accounted for in the conclusion [32]. CASP confirms PEDro’s findings regarding randomisation and the presence of blind assessors [29,30,33,34]. In two studies, it is unclear whether the groups were similar at baseline due to substantial differences in sample sizes between groups [30,34]. All studies found that TPR provides equivalent or greater value to people in care than any other existing intervention. Table 5 presents all the specifications.

### 3.7. Certainty of Evidence Analysis

The certainty of evidence for the outcomes was carefully evaluated. Adherence outcomes were rated as low certainty for both comparisons (TPR versus PR and TPR versus usual care) owing to a significant risk of bias and inconsistency. Dropout results were assigned moderate certainty since they revealed fewer methodological concerns. Quality-of-life outcomes were also rated as low certainty, primarily due to a serious risk of bias and inconsistency in both comparisons. Consequently, these findings should be interpreted with caution, and further high-quality studies are necessary to confirm the results. GRADE results are presented in Table 6.

## 4. Discussion

This systematic review analysed patient adherence, satisfaction, and quality-of-life (QoL) improvement in Chronic Obstructive Pulmonary Disease (COPD) patients who underwent telerehabilitation (TPR). After reviewing eight RCTs, we observed considerable variability in patient adherence to TPR compared to more stable pulmonary rehabilitation (PR) adherence. High patient satisfaction was reported across the different interventions. Among the studies, dropout rates varied; some demonstrated higher dropout in the PR or usual care group, while others had higher dropout in the TPR group. Several QoL scales were utilised; the majority of the research reported no significant difference in QoL improvements between TPR and PR. However, significant improvements were observed when TPR was compared to usual care. The studies showed substantial differences in the adherence evaluation method, TPR intervention features (frequency/delivery/materials), diversity in satisfaction questionnaires and QoL scales, and dropout rate variability. This could explain the studies’ heterogeneous results. The Discussion will cover the primary, secondary, and tertiary outcomes individually.

### 4.1. Primary Outcomes

Currently, few Randomised Controlled Trials (RCTs) have focused on monitoring patient adherence to telerehabilitation in COPD patients. Assessing adherence is challenging and often inadequately reported in studies due to varied measurement methods and tracking difficulties, which complicate comparisons between studies [36]. No reference values exist to classify adherence. Adherence was classified in a few studies, with participants attending at least 70% of prescribed sessions [33,37,38]. One study [33] found a higher adherence rate to TPR compared to PR, although this was not considered significant. This highlights the challenges of adherence in conventional programs, which TPR can overcome. Other articles in the literature showed higher adherence to TPR [18,38], with Cox et al. [38] demonstrating TPR adherence of 84% of ≥70% of prescribed sessions, classified as high adherence compared to 79% for PR. Chaplin et al. [39] and Holland et al. [18] explored telerehabilitation in COPD patients, which aligned with our study’s outcomes, but were not included in our search strategy. Holland et al. [18] defined adherence as previously stated, with a 49% completion rate in PR versus 91% in TPR, explained by a highly structured TPR program and ease of use, confirming accessibility for attendees. Three selected articles [29,32,35] showed TPR adherence rates to exercises superior to 80%, consistent with an external article reporting 91% adherence [18]. This is significantly higher than the lowest TPR adherence rates found in this systematic review, which were 21% [28] and 59% [34], or 56.2% in an external study [13].

Many factors influence adherence to a rehabilitation program. One systematic review [40] found a fair association between higher adherence and more frequently prescribed exercises. This could explain the lower adherence rates found in Tabak et al. [28] and Polo et al. [34], as they have fewer sessions than other analysed TPR programs. Patients’ motivation and preferences regarding the type of rehabilitation (in-person or online) may influence the program’s outcomes [18], as confirmed by external authors Chaplin et al. [39], with some participants preferring TPR over PR. However, an individual’s preference for a program does not guarantee its completion. According to Stampa et al. (2024) [41], technical skills and issues with TPR delivery are identified as barriers, potentially explaining the adherence rates related to equipment difficulties [34] and activity coach frustration [28]. Patient motivation is considered an important facilitator [41]; Hansen et al. [33], while finding no significant between-group differences in adherence, demonstrated lower adherence to PR programs and higher dropout rates, the most reported reason being the lack of motivation. This confirms the lower motivation to participate in PR, explaining the slightly better adherence and lower dropout rates in TPR. Low adherence to TPR exercises [28] could be attributed to physiotherapists’ lack of regularity in exercise prescriptions, probably due to a distrust of online rehabilitation technology, which influences patient adherence to a program. Bourne et al. [30] showed differences in session frequency achieved between groups (TPR: five suggested sessions versus two attended for PR) and sample size disparity, potentially impacting the consistency of adherence rates and explaining the lower adherence rate in the TPR group.

The lack of clear consensus on what might be considered acceptable adherence in clinical trials makes it difficult to draw consistent conclusions for our systematic review [16]. According to the WHO [13], the average adherence to long-term therapy in chronic disease is 50%; all selected articles exceed this, suggesting acceptable adherence, except for one [28]. A recent systematic review [16] established 80% or more of prescribed exercises as the threshold for satisfactory adherence to TPR in the musculoskeletal disorders population, showing satisfactory TPR exercise adherence for three selected articles [29,32,35]. Betancourt-Peña et al. [42] aimed to determine adherence to PR in COPD patients. Adherence was classified by the number of sessions completed: less than 35% corresponds to low adherence, 35–85% to moderate adherence, and more than 85% to high PR adherence. By comparing to TPR, two studies had high adherence rates, 93.5% [32] and a mean adherence of 91.7% (mean 22/24 sessions) [29]. Cerdán-De-Las-Heras et al. [35] showed high adherence at 3 (90%) and 6 months’ (85%) follow-up and moderate adherence at 8 weeks (82%). Galdiz et al. [31] also demonstrated moderate exercise adherence over a 365-day TPR program duration (60%). This can be seen as a positive indicator, showing that programs of longer duration did not result in significantly lower adherence rates, confirmed by the results of Bourne et al. [30], which show a similar adherence rate (62%, moderate) despite a much shorter study duration (42 days) [40]. Polo et al. [34] and Hansen et al. [33] reported moderate adherence (59%/73%, respectively). Tabak et al. [28] reported a low adherence rate to exercises (21%) while reporting high adherence (86.4%) to the online diary.

### 4.2. Secondary Outcome

TPR programs appear to have a higher retention rate than PR programs. High dropout rates are often observed in conventional PR [13,39,43]. Four out of eight studies indicated higher PR/usual care dropout rates compared to TPR [28,31,33,35]. Yohannes et al. (2022) confirmed that 25% of patients did not complete the PR program [44]. Hansen et al. [33] reported a 32% dropout rate for PR, primarily due to a lack of motivation, a finding supported by Pacheco et al. [45], who stated that the main reasons for PR program dropouts are related to poor motivation. In one study [30], TPR had a higher dropout rate (28%) compared to PR (20%), which can be attributed to the study itself, due to the absence of an intensive online protocol or phone mentoring. Alhasani et al. [36] confirmed that interventions provided by healthcare professionals, including strategies such as motivational messages, can reduce dropout rates and enhance patient participation and motivation. The materials used to deliver TPR also play a crucial role; according to Ringbaek et al. [46], the tablet group experienced a lower dropout rate. This was corroborated by Cerdán- De- Las- Heras et al. [35], who utilised tablets for TPR and observed a lower dropout rate in the TPR group. Studies focusing on patient satisfaction [28,34,35] employed various questionnaires to assess it, which may lead to inconsistencies. Cerdán- De- Las- Heras et al. [35] found that a majority of patients were satisfied with TPR, but there was no information regarding patient satisfaction in PR, which limits the interpretation of results. Generally, patients express high satisfaction with both TPR and PR or usual care, as shown in our three included studies [28,34,35] and corroborated by other research [37,46]. High patient satisfaction may contribute to higher patient adherence to programs [35]. Polo et al. found no difference between the two groups, though there was a noteworthy difference regarding patients who were very satisfied with the support provided by the TPR team. Conversely, Tabak et al. found slightly lower patient satisfaction in the TPR group, attributed to frustration with activity coaches due to participants’ preferences for specific types of exercise.

### 4.3. Tertiary Outcomes

Strong patient adherence could be linked to improvements in a patient’s QoL. Vasilopoulou et al. [32] achieved 93.5% adherence and reported that TPR was equally effective as PR in maintaining clinically meaningful QoL improvements over usual care, confirming the link (adherence/QoL). Furthermore, Galdiz et al. [31] demonstrated moderate adherence to TPR exercises based on prior interpretation [42], showing QoL improvements favouring TPR over usual care. Tabak et al. [28] reported high patient adherence to the web portal, accompanied by QoL improvements; however, the missing *p*-value prevents this difference from being considered statistically significant. The COPD Assessment Test (CAT), Clinical COPD Questionnaire (CCQ), and modified Medical Research Council (mMRC) are the only QoL scales specifically designed for the COPD population, whereas St. George’s Respiratory Questionnaire (SGRQ), Euro-QoL-5, Chronic Respiratory Disease Questionnaire (CRQ), and SF-36 are more generic scales that may introduce bias, leading to inconsistent measurements. The findings indicated that TPR was equally effective as PR concerning QoL improvements [30,33,34,35] and that both TPR and PR provide significant QoL improvements over usual care [29,31,32].

### 4.4. Limitations

This study was limited by the paucity of RCTs on the subject, resulting in a lack of clear consensus on what can be regarded as acceptable adherence due to the absence of established reference values for classifying adherence to rehabilitation programs. The analysis of adherence to TPR compared to PR and/or usual care was complicated by the fact that four articles did not calculate adherence to PR/usual care [28,29,32,35]. All the articles measured adherence in different ways and used various quality-of-life scales, impacting the consistency of results. In addition, despite the good quality of the evidence, one study [34] may have influenced the study results due to inconsistencies in data reporting. A meta-analysis was not feasible due to substantial heterogeneity across studies. Adherence was measured and reported differently in each study, using various definitions and tools, with inconsistent or missing methodological details. This prevented the standardisation of outcomes necessary for data pooling. Therefore, we followed the Synthesis Without Meta-analysis (SWiM) guidelines and conducted a narrative synthesis instead. Furthermore, publication bias may lead to overestimating adherence because studies with low adherence are less often published, biasing the literature toward better adherence outcomes. Language bias, by including only studies in English, French, or Spanish, excludes research from other languages. All these biases limit the generalisability of the results.

### 4.5. Future Perspective

Some gaps remain to be addressed, notably the need to create a consensus core outcome set for adherence to standardise its definition and measurement. The development of automated adherence tracking through digital devices could provide more objective and consistent data. Conducting more high-quality studies with these aims will help create reference values for classifying adherence, facilitating more reliable comparisons across studies.

## 5. Conclusions

The findings of this systematic review reveal heterogeneity in adherence classification, leading to an unclear level of adherence to TPR compared to PR and/or usual care. In fact, in four selected articles, adherence to TPR appears equivalent or superior to PR and usual care. Furthermore, TPR programs frequently recorded lower dropout rates than PR programs and usual care. Patients were satisfied with TPR, and greater satisfaction tends to lead to better adherence to a program. Regarding QoL improvements, TPR was equally successful as PR, and both were superior to usual care. However, the observed heterogeneity and inconsistent definitions of adherence preclude firm conclusions, highlighting the need for research into homogeneous criteria to define adherence. This will guide a definitive conclusion about whether patient adherence to TPR is similar to conventional therapy and usual care.

## Figures and Tables

**Figure 1 healthcare-13-01818-f001:**
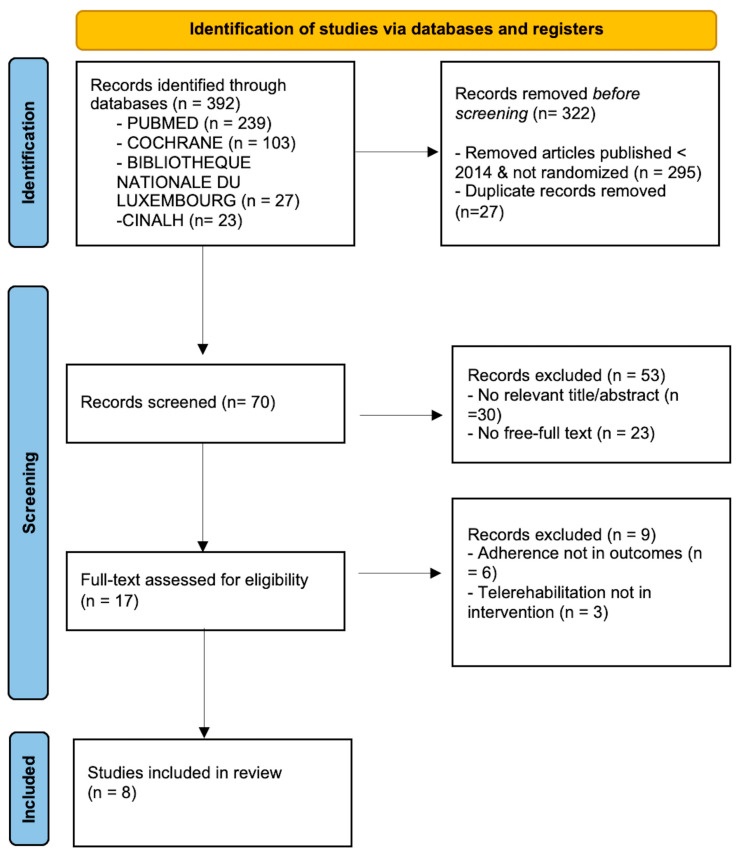
PRISMA search strategy flow diagram [27].

**Table 1 healthcare-13-01818-t001:** Baseline study characteristics.

Author, Year, Country	Sample (n)		Gender (F)		Age (Years) *		COPD Severity (FEV1% Predicted) *		mMRC or MRC, *		CAT, *		SGRQ, *		Euro-Qol 5-Dimension, * (VAS/Index)		CRQ (D/F/E/M), Mean (SD)		SF-36 (PCS/MCS, Mean (SD))		CCQ (Mean ± (SD))	Programs Duration	Number of Sessions/ Exercises/ Exercise Times Attended	
	TPR	PR/UC	TPR	PR/UC	TPR	PR/UC	TPR	PR/UC	TPR	PR/UC	TPR	PR/UC	TPR	PR/UC	TPR	PR/UC	TPR	PR/UC	TPR	PR/UC	TPR/ PR or Usual Care	TPR/PR/Usual Care	TPR	PR/UC
Bourne, 2017, United Kingdom [30]	n = 64	n = 26	n = 23	n = 8	69.1 ± 7.9	71.4 ± 8.6	58.0 ± 23.6	60.5 ± 20.1	mMRC: 2.0 (1.0–3.0)	mMRC: 2.0 (1.0–2.0)	18.1 ± 7.9	17.3 ± 6.7	42.4 ± 18.6	37.7 ± 17.2	NR	NR	NR	NR	NR	NR	NR	42 days	Week 1/2/3/4/5/6 (mean) 3.9/3.5/3.2/3.0/2.8/2.5	Week 1/2/3/4/5/6 (mean) 1.6/1.3/1.5/1.5/1.3/1.4
Cerdán-de-las-Heras, 2021, [35] Denmark	n = 27	n = 27	n = 11	n = 12	67.4 ± 10.2	72.5 ± 7.4	36.1 ± 14.1	32.8 ± 8.5	NR	NR	NR	NR	55.6 ± 13.5	60.6 ± 14.1	NR	NR	NR	NR	NR	NR	NR	56 days	Exercise minutes expected 480/ trained 394 (0–8 weeks)	NR
Galdiz, 2021, [31] Spain	n = 46	UC n= 48	n ≃ 16 (34.8%)	n ≃ 15 (31.2%)	62.3 ± 8.2	63.0 ± 6.6	42.09 ± 14.59	45.87 ± 17.24	mMRC: 2.00 (0.92)	mMRC: 1.78 (0.77)	16.02 ± 6.08	15.87 ± 7.83	NR	NR	NR	NR	5.2 (1.2)/4.7 (1.10)/5.1 (1.2)/5.2 (1.5)	5.2 (1.5)/ 4.8 (1.3)/ 5.2 (1.3)/ 5.3 (1.5)	52.0 (20.2)/ 71.0 (18.8)	54.0 (22.7)/ 75.4 (18.4)	NR	365 days	Compliance to scheduled exercises: 60%	NR
Hansen, 2020, [33] Denmark	n = 67	n = 67	n = 35	n = 39	68.4 ± 8.7	68.2± 9.4	32.6 ± 10.3	33.7 ± 8.4	MRC: 0/2/30/27/8	MRC: 0/0/35/23/9	19.8 ± 7.3	20.4 ± 6.6	NR	NR	51.5 ± 19.4/0.66 ± 0.20	53.9 ± 19.1/ 0.70 ± 0.12	NR	NR	NR	NR	2.7 (0.9)/ 2.9 (1.0)	70 days TPR, 70 days for 7 PR, 84 days for 1 PR program	Median: 25 sessions	Median: 16 sessions
Polo, 2023, [34] United States	IDR: n = 57	IDR: n = 28	n = 37	n = 18	67.35 ± 12.05	66.50 ± 9.96	51 ± 27	48 ± 19	mMRC TPR (n = 62): 2.46 ± 1.16	mMRC PR (n = 37): 2.24 ± 1.04	TPR (n = 62): 22.37 ± 8.35	PR (n = 37): 20.89 ± 8.53	NR	NR	NR	NR	NR	NR	NR	NR	NR	56 days	TPR (n = 57): mean sessions (SD): 9.47/16 (4.46)	PR (n = 28): mean sessions (SD): 10.07/16 (5.72)
Tabak, 2014, [28] Netherlands	n = 12	UC n = 12	n = 6	n = 6	64.1 ± 9.0	62.8 ± 7.4	50.0	36.0	MRC: 3.0	MRC: 4.0	NR	NR	NR	NR	64.1 ± 4.9/0.75 ± 0.09	65.0 ± 12/ 0.69 ± 0.14	NR	NR	NR	NR	2.0 ± 1.0/ 2.7 ± 0.8	273 days	569 exercises prescribed, 127 performed	NR
Tsai, 2017, [29] Australia	n = 19	UC n = 17	n = 7	n = 11	73 ± 8.0	75 ± 9.0	60 ± 23	68 ± 19	mMRC: 2.00 (1)	mMRC: 2.00 (1)	16 ± 7	15 ± 6	NR	NR	NR	NR	17(7)/ 15(5)/ 35(9)/ 22(4), total score 90 (18)	18 (8)/ 15 (7)/ 33 (9)/ 21 (5), total score 88 (23)	NR	NR	NR	56 days	Mean ± SD: 22 ± 5/24 sessions	NR
Vasilopoulou, 2017, [32] Greece	n = 47	n = 50/UC = 50	n = 3	n = 12/ n = 13	66.9 ± 9.6	66.7 ± 7.3/ 64.0 ± 8.0	49.6 ± 21.9	51.8 ± 17.3/ 51.7 ± 21.0	mMRC: 2.3 ± 1.0	mMRC: 2.5 ± 1.0/ (2.2 ± 1.1)	17.6 ± 8.1	15.7 ± 5.6/ 15.8 ± 4.9	46.2 ± 19.7	43.5 ± 16.7/ 44.1 ± 16.6	NR	NR	NR	NR	NR	NR	NR	365 days	Adherence: 93.5%	Adherence to training sessions: 91%

IDR: inconsistencies in data report; UC: usual care; standard deviation (SD), Global Initiative for Chronic Obstructive Lung Disease (GOLD), Forced Expiratory Volume in 1 s (FEV1), number of participants (n=), not reported (NR), Visual Analog Scale (VAS), modified Medical Research Council (mMRC), Medical Research Council (MRC), COPD Assessment Test (CAT), St. George’s Respiratory Questionnaire (SGRQ), Chronic Respiratory * Mean ± SD; Questionnaire Dyspnea, Fatigue, Emotion, Mastery (CRQ-D/F/E/M), Short-Form 36 Physical Component Summary, Mental Component Summary (SF-36 PCS/MCS), Clinical COPD Questionnaire (CCQ).

**Table 2 healthcare-13-01818-t002:** Intervention characteristics.

Author, Year	TPR Program	Materials/Delivery Mode	TPR Session Frequency and Duration	PR/Usual Care Program	PR Session Frequency and Duration
Bourne, 2017 [30]	10 video-facilitated exercises: warm-up/biceps curl/squat/push-up against wall/leg extension/upright row with weight/sit to stand/arm swing with a stick/leg kicks/arm punches with weights/step-ups/cool-down/educational sessions (anatomy, disease explanations, self-management, breathing technique…)	Not reported/online via log into myPR, no information on electronic device used (“on-screen”)	10 exercises 2 to 5×/week lasting 60 s per exercise (increase by 30 s each week) combined with 3×/week educational sessions for 6 weeks	Same program as TPR	2× supervised and 3× home combined with 3× educational sessions per week for 6 weeks
Cerdán-de-las-Heras, 2021 [35]	Video consultations, e-learning package (psychological, medical, nutritional, physical aspects of COPD), physical training, online questionnaires, chat sessions/tailored aerobic and anaerobic workout	Weight, elastic, fitness step/mobile VAPA app on smartphone or tablet	10–20′ 3–5×/week for 8 weeks	Conventional rehabilitation program	2×/per week 1 h and 6 h of educational sessions in total for 8 weeks
Galdiz, 2021 [31]	Video-facilitated exercises: 10 min chest physiotherapy, 30 min arm weight lifting/30 min leg cycle ergometry	Pulse ergometer–dumbbells–exercise bicycle/web-based platform on mobile phone	Approximately 1 h and half, at least 3×/week for 12 months	Advice to exercise regularly, walking 1 h daily	Maintenance program
Hansen, 2020 [33]	Supervised real-time sessions: warm-up (5 min), 35 min exercises 50/50% exercises for upper and lower extremities (sit to stand, biceps curls, shoulder press, step-up, bent over rowing, static dynamic squat, front raise dumbbells)/20 min educational sessions (advices: nutrition, smoking, COPD and anxiety management, use of respiratory devices…)	One step-box and dumbbells (pairs of 1–10 kg)/video conference software system, single touch screen	60 min (exercise + educational sessions), 3×/week for 10 weeks	Warm-up (10 min), endurance 20–30 min: cycle or walking or treadmill or circuit training or activity games/resistance training 20–30 min: 50/50% upper and lower limb (leg press, knee extension, pull down and/or chest press)/cool-down 5–10 min: breathing exercises, yoga/educational sessions (importance of nutrition, smoking cessation, daily exercises, use of respiratory device…)	Exercise sessions 60 min 2×/per week and 1×/per week educational sessions (60–90 min) for 10 weeks for 7 PR hospitals and 12 weeks for 1 PR hospital program
Polo, 2023 [34]	Similar to conventional rehabilitation: supervised real-time sessions (up to 3 participants) 30 min cycle ergometer, 20 min anaerobic exercise, 10 min cool-down/educational videos (physical and breathing exercises, medications management…)	Bicycle, weights, stretch bands, vital sign monitor (watch)/tablet	60 min 2×/week for 8 weeks	30 min of aerobic exercise on a treadmill, 20′ anaerobic exercise, 10′ cool-down + educational videos	60 min 2×/week for 8 weeks
Tabak, 2014 [28]	Video-facilitated resistance and endurance exercises, breathing exercises, relaxation, mobilisation, mucus clearance, two 90 min educational sessions attended before program start (early symptoms detection before exacerbation)	Accelerometer-based activity sensor and smartphone for activity coach module, did not report materials for exercises/web portal, no information on electronic device used for TPR	Not reported, no standardised protocols, 569 exercises are expected to be executed within 9 months	Usual care, self-management, physiotherapy sessions (if prescribed)	Maintenance program
Tsai, 2017 [29]	Supervised group exercise via real-time videoconferencing, warm-up (5 min cycle ergometer), 15–20 min lower limb cycle ergometery, 15–20 min walking training and strengthening exercises (3 × 10 repetitions sit to stand and squats)	Laptop computer, lower limb cycle ergometer, pulse oximeter/computer with an in-built camera	Approximately 1 h, 3×/week for 8 weeks	Usual medical management (optimal pharmacological intervention) + action plan, no exercise training	Maintenance program
Vasilopoulou, 2017 [32]	Video-demonstrated arm and leg exercises/walking drills/patient education (breathing retraining techniques, dietary, self-management advice, psychological support…)	Spirodoc, pedometer, oximeter, did not report materials for exercises/tablet (video and data recording)	1 h session 3×/week for 12 months	Similar to primary 2-month PR program: aerobic (45 min) and resistance exercises (15 min), patient education (dietary advice, self-management, psychological advice and breathing technique)/usual care self-management	2× per week for 12 months (similar to primary 2-month PR: approximately 1 h)/maintenance program

Virtual-Autonomous-Physiotherapist-Agent (VAPA).

**Table 3 healthcare-13-01818-t003:** Outcomes and results.

First Author, Year	Primary Outcomes	Results	Secondary Outcomes	Results	Tertiary Outcomes	Results
Bourne, 2017 [30]	Treatment adherence and compliance: Mean of the number of exercise sessions completed.	**TPR:** 62% of the five suggested sessions attended/decrease in participation: from 3.9 (week 1) to 2.5 (week 6) mean sessions per participant completed	*Dropouts*	**TPR:** 15 participants/64 (23%) withdrew (11) and were lost to follow-up (4) + 3 additional had exacerbations	*Quality of life:* CAT, SGRQ, mMRC	CAT score difference in ITT was −1.0 in favour to TPR but no clinically important minimum difference, mMRC and SGRQ suggested non-inferiority for TPR.
		**PR:** 72% of the two sessions attended/stable participation: from 1.6 (week 1) to 1.4 (week 6) mean sessions per participant completed		**PR:** 19% withdrew (3) or were lost to follow-up (2) = (5 participants/26)		
Cerdán-de-las-Heras, 2021 [35]	Treatment adherence and compliance: Duration of exercise set performed and weekly average training time based on expected exercising time (minimum 60 min/week).	**TPR:** 82% of adherence (0–8 W), 90% of adherence (8W-3M), 85% of adherence (3–6M)/increase in exercise set mean time (+25% 0–8W/+66% 8W-6M) and training time performed (from 0–8W: 394 to 3–6M: 614 min)	*Patient satisfaction (5-point Likert scale, 1—very unsatisfied, 5—very satisfied)/dropouts*	**TPR:** 5-point Likert scale scored 4.27/dropouts = 12 (provide reasons without specifying which groups)	*Quality of life:* SGRQ	TPR vs. PR show no difference in improvement of QoL based on SGRQ.
		**PR:** No information reported		**PR:** No information reported for satisfaction/dropouts = 19		
Galdiz, 2021 [31]	Treatment adherence and compliance: **Adherence =** number of appointments, **compliance =** number of sessions completed based on scheduled one. Non-complier = not perform exercises for 8 consecutive weeks/non-adherent = not attend 2 follow-up appointment.	**TPR:** 92.4% of patients attended scheduled appointments (4 patients are non-adherent)/60% of compliance for scheduled exercise days (12 patients are non-complier)	*Dropouts*	**TPR:** 2 not meeting inclusion criteria, 2 voluntary withdrawal, 1 severe exacerbation	*Quality of life:* mMRC, CAT, CRQ, SF-36	TPR shows both significant and clinical improvements for SF-36 MCS and CRQ-E compared to usual care. Significant different time profiles between groups mainly at 3 months for CRQ-D and CRQ-M in favour of TPR.
		**Usual care:** 84.4% of patients attended scheduled appointments (4 non-adherent patients)		**Usual care:** 1 not meeting inclusion criteria, 2 voluntary withdrawal, 5 excluded from analysis (no 12-month results)		
Hansen, 2020 [33]	Treatment adherence and compliance: Participation in entire scheduled session *(exercise/education)*.	**TPR:** median of 25 sessions/73% of patients attended ≥70% of programs’ total sessions (49/67 sessions)	*Dropouts*	**TPR:** 1 dead, 2 illness, 1 exacerbation, 4 low motivation, 2 private issues= 10 dropouts	*Quality of life:* MRC, CAT, Euro-QoL-5	CAT shows statistical difference between groups in symptoms reduction for TPR but not exceeding MCID. No significant difference between TPR and PR according to MCID for Euro-QoL-5.
		**PR:** median of 16 sessions/62% of patients attended ≥70% of programs’ total sessions (42/67 sessions)		**PR:** 2 dead, 2 adverse events, 6 illness, 4 exacerbations, 9 low motivation, 1 private issue= 24 dropouts		
Polo, 2023 [34]	Treatment adherence and compliance: Number of sessions attended out of the total of 16 sessions.	**TPR:** 9.47/16 sessions attended: average of 59.21% of sessions completed/47/57 participants completed the 8-week program	*Patient satisfaction (Satisfaction Survey Questions)/dropouts*	No significant differences in satisfaction survey between TPR and PR, only significant for the question of the perceived usefulness of the team (92.86% (TPR) vs. 81.48% (PR) *p*-value = 0.044); both groups were very satisfied	*Quality of life:* mMRC, CAT	TPR shows improvement for CAT and mMRC from day 1 to 8 weeks but regressed at 12 months follow-up, TPR is equivalent to PR for QoL.
				**TPR:** 2 withdrew (1 died), 1 amputation		
		**PR:** 10.07/16 sessions attended: average of 62.95% of sessions completed/20/28 participants completed the 8-week program		**PR:** 3 unable to complete (1 died)		
Tabak, 2014 [28]	Treatment adherence and compliance: **Adherence to online diary** = number of diary fill-outs divided by number of treatment days. **Adherence to exercises** = number of exercises prescribed divided by number performed.	**TPR:** adherence to online diary: web portal was used in 86.4% of days during intervention	*Patient satisfaction (Client satisfaction-8 questionnaire, highest score (8–32) = higher patient satisfaction)/dropouts*	Good satisfaction for TPR and usual care (1 month: mean 26.4/30.4, 3 months: mean 26.3/29.9, respectively) = slightly lower TPR satisfaction due to activity coach frustration	*Quality of life:* MRC, Euro-QoL-5	QoL (Euro-QoL-5) improved for TPR and declined for usual care, but significant difference is unknown (missing *p*-value).
		**TPR:** adherence to exercises: on 569 exercises prescribed, 127 were completely performed. Low adherence to exercises (21%)		**TPR:** 1 personal worry, 1 exacerbation		
		**Usual care:** No information reported		**Usual care:** 2 “too much effort”, 2 exacerbations, 2 hospitalisations, 1 technical issue, 1 kidney problem, 2 “rehabilitation program”		
Tsai, 2017 [29]	Treatment adherence and compliance: Number of completed exercise training sessions based on a possible 24 sessions.	**TPR:** high adherence, mean ± SD: 22 ± 5 of sessions performed (91.7%)	*Dropouts*	**TPR:** 1 loss to follow-up, (death but not related to the study)	*Quality of life:* CAT, CRQ	Within-group: significant improvement in CRQ total score for TPR, between-group: CRQ total score trend toward significant improvement of 8 points for TPR (*p*-value = 0.07) over usual care/CAT: no statistically significant between-group differences.
		**Usual care:** No information reported		**Usual care:** No dropout		
Vasilopoulou, 2017 [32]	Treatment adherence and compliance: **Adherence to exercises** = number of sessions performed based on total number of expected sessions. **Adherence to measurements** = number of registrations entered divided by the number of those recommended.	**TPR:** good overall adherence of 93.5%, adherence to home exercises of 91.7%	*Dropouts*	No dropouts over 12-month period between groups	*Quality of life:* mMRC, CAT, SGRQ	TPR was equally effective as PR in maintaining initial clinically meaningful improvement in SGRQ, CAT, and mMRC scores over 12-month period, superior to usual care.
		**PR:** good adherence to exercise training sessions of 91%				
		**Usual care:** No information reported				

Analog Scale (VAS), modified Medical Research Council (mMRC), Medical Research Council (MRC), COPD Assessment Test (CAT), St. George’s Respiratory Questionnaire (SGRQ), Chronic Respiratory Questionnaire Dyspnea, Fatigue, Emotion, Mastery (CRQ-D/F/E/M), Short-Form 36 Physical Component Summary, Mental Component Summary (SF-36 PCS/MCS), Clinical COPD Questionnaire (CCQ).

**Table 4 healthcare-13-01818-t004:** PEDro scales [24].

	PEDro Scores												
Author, Year	EC	RA	CA	BC	BS	BT	BA	Follow-Up	ITA	BGC	Point Estimates/Variability	Total Score	Quality of Evidence
Bourne, 2017 [30]	YES											8	Good

Cerdán-de-las-Heras, 2021 [35]	YES											4	Fair

Galdiz, 2021 [31]	YES											7	Good

Hansen, 2020 [33]	YES											7	Good

Polo, 2023 [34]	YES											6	Good

Tabak, 2014 [28]	YES											6	Good

Tsai, 2017 [29]	YES											8	Good

Vasilopoulou, 2017 [32]	YES											5	Fair

Green (yes)/red (no); EC (eligibility criteria); RA (random allocation); CA (concealed allocation); BC (baseline comparability); BS (blind subjects); BT (blind therapists); BA (blind assessors); ITA (intention-to-treat analysis); BGC (between-group comparisons).

**Table 5 healthcare-13-01818-t005:** CASP checklist analysis [2].

CASP Question Number	Authors							
	Bourne et al. [30]	Cerdán-de-las-Heras et al. [35]	Galdiz et al. [31]	Hansen et al. [33]	Polo et al. [34]	Tabak et al. [28]	Tsai et al. [29]	Vasilopoulou et al. [32]
1. Did the study address a clearly focused research question?	YES	YES	YES	YES	YES	YES	YES	YES
2. Was the assignment of participants to interventions randomised?	YES	YES	YES	YES	YES	YES	YES	YES
3. Were all participants who entered the study accounted for at its conclusion?	YES	YES	YES	YES	YES	YES	YES	NO
4. Were patients/health workers/study personnel “blind” to treatment?	NO/NO/YES	NO	NO	NO/NO/YES	NO/NO/YES	NO	NO/NO/YES	NO
6. Apart from the experimental intervention, were the groups treated equally?	YES	YES	YES	YES	YES	YES	YES	NO
7. Were the effects of intervention reported comprehensively? *	YES	YES	YES	YES	YES	YES	YES	YES
8. Was the precision of the estimate of the intervention or treatment effect reported? **	YES	YES	YES	YES	YES	YES	YES	YES
9. Are the benefits worth the harms and costs?	Cannot Tell	Cannot Tell	Cannot Tell	Cannot Tell	Cannot Tell	Cannot Tell	Cannot Tell	Cannot Tell
10. Can the results be applied to your local population/in your context?	YES	YES	YES	YES	YES	YES	YES	YES
11. Would the experimental intervention provide greater value to the people in your care than any of the existing interventions?	YES	YES	YES	YES	YES	YES	YES	YES

* Results must be clearly stated based on sample size power calculation and primary results. ** Based on the extract *p*-value and confidence interval value of primary results.

**Table 6 healthcare-13-01818-t006:** GRADE [27].

Number of Studies	Comparison	Risk of Bias	Inconsistency	Indirectness	Imprecision	Publication Bias	Intervention (n)	Comparator (n)	Certainty
		Adherence							
5 RCTs: Bourne et al. [30], Cerdán-De-Las-Heras et al. [35], Hansen et al. [33], Polo et al. [34], Vasilopoulou et al. [32]	TPR vs. PR	Serious (a)	Serious (b)	Not serious	Not serious	Undetected	262	198	⊕⊕OO Low
4 RCTs: Galdiz et al. [31], Tabak et al. [28], Vasilopoulou et al. [32], Tsai et al. [29]	TPR vs. usual care	Not serious	Serious (b)	Not serious	Serious (d)	Undetected	125	127	⊕⊕OO Low
		Drop-out							
5 RCTs: Bourne et al. [30], Cerdán-De-Las-Heras et al. [35], Hansen et al. [33], Polo et al. [34], Vasilopoulou et al. [32]	TPR vs. PR	Serious (a)	Not serious	Not serious	Not serious	Undetected	262	198	⊕⊕⊕O Moderate
4 RCTs: Galdiz et al. [31], Tabak et al. [28], Vasilopoulou et al. [32], Tsai et al. [29]	TPR vs. usual care	Not serious	Not serious	Not serious	Serious (d)	Undetected	125	127	⊕⊕⊕O Moderate
		Quality of life							
5 RCTs: Bourne et al. [30], Cerdán-De-Las-Heras et al. [35], Hansen et al. [33], Polo et al. [34], Vasilopoulou et al. [32]	TPR vs. PR	Serious (a)	Serious (c)	Not serious	Not serious	Undetected	262	198	⊕⊕OO Low
4 RCTs: Galdiz et al. [31], Tabak et al. [28], Vasilopoulou et al. [32], Tsai et al. [29]	TPR vs. usual care	Not serious	Serious (c)	Not serious	Serious (d)	Undetected	125	127	⊕⊕OO Low

a. Polo et al. reported a number of participants with inconsistency. b. All authors used different methods to quantify adherence, except for three [29,33,35], leading to inconsistency (consult Table 3). c. Quality of life is not reported using the same scales between studies (refer to Table 1). d. Fewer than 300 participants ⊕⊕⊕O represents moderate certainty of evidence, ⊕⊕OO signifies low certainty.

## Data Availability

No new data were created or analysed in this study.

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
