# Peer review of "Pulmonary Telerehabilitation in COPD Patients: A Systematic Review to Analyse Patients’ Adherence"

_healthcare, 2025, doi:10.3390/healthcare13151818_

Round 1
Reviewer 1 Report
Comments and Suggestions for Authors
This study aimed to investigate the adherence of COPD patients to TPR. I have the following comments:
-
The authors reported heterogeneity in how adherence is measured across studies. A table categorizing adherence definitions (session attendance, exercise duration, percentage of completed activities) should be added. The implications of these varying definitions should also be discussed.
-
The review presents a purely narrative synthesis. The authors should consider whether a meta-analysis could be conducted. If not feasible, a clear justification should be provided.
- The inclusion of only RCTs should be discussed. Are there any non-RCTs?
- The GRADE summary table should be described in more detail in the main text. Which outcomes had low vs. high certainty, and how does that affect the authors' conclusions?
- All abbreviations should be defined on first use.
Author Response
Thank you very much for reviewing our thesis and for your insightful and constructive comments. We greatly appreciate your feedback, which has been extremely helpful in improving the quality of our work.
We have carefully considered all your suggestions and have made every effort to incorporate them into the revised version of the thesis. Your input has been invaluable in refining our arguments and improving the clarity of our analysis.
Reviewers 1
- The authors reported heterogeneity in how adherence is measured across studies. A table categorizing adherence definitions (session attendance, exercise duration, percentage of completed activities) should be added. The implications of these varying definitions shouldalso be discussed.
Answer: Thank you for your comment. We would like to clarify that all definitions of adherence have already been included in Table 3. Consequently, we did not create a separate table specifically for adherence, as we chose to consolidate all our outcomes into a single comprehensive table. We believe this better represents the characteristics of the included studies.
- The review presents a purely narrative synthesis. The authors should consider whether a meta-analysis could be conducted. If not feasible, a clear justification should be provided.
Answer: We have revised the manuscript and included a clear justification in the limitations section explaining why a meta-analysis was not feasible.
- The inclusion of only RCTs should be discussed. Are there any non-RCTs?
Answer: Only RCTs were included; there were no non-RCTs in our selection, This is issue was highlighted in the methods.
- The GRADE summary table should be described in more detail in the main text. Which outcomes had low vs. high certainty, and how does that affect the authors' conclusions?
- All abbreviations should be defined on first use.
Answer: thank foir your comment, corrections were done.
Reviewer 2 Report
Comments and Suggestions for Authors
General appraisal
Your topic is timely and clinically relevant: telerehabilitation offers a scalable way to expand pulmonary-rehabilitation (PR) benefits to people with COPD, and adherence is indeed the hinge outcome. The review is potentially publishable, but several issues undermine its clarity, reproducibility and scientific rigour. My comments focus on concrete, inside-the-scope revisions Please read them as collegial suggestions aimed at strengthening the manuscript.
1 Introduction
Issue |
Actionable recommendation |
1.1 Conceptual framing – Adherence is introduced but not defined; you later lament the absence of a universal definition. |
Add an operational definition up-front (e.g., WHO, ISPOR or ATS) and state why heterogeneity hampers comparisons. |
1.2 Redundancy / length – Multiple sentences repeat global COPD burden figures; section spans >650 words. |
Compress global-burden statistics to one sentence; dedicate space to conceptual gaps (e.g., why previous reviews did not resolve adherence heterogeneity). |
1.3 Citations – Some references are duplicated (e.g., refs 16/41) or missing DOIs. |
De-duplicate and cross-check each citation; supply DOIs per journal style. |
1.4 Rationale / objectives – Objective blends adherence, satisfaction, QoL but does not signal hierarchy. |
End with a clear, bullet-style aim: “Primary: quantify adherence; Secondary: compare drop-out and satisfaction; Tertiary: examine QoL.” |
2 Methods
Issue |
Recommendation |
2.1 Search strategy transparency – The string is paraphrased and the Boolean logic is unclear. |
Provide the full PubMed strategy (search line-by-line, with date of final run) in Supplement S1; replicate for other databases. |
2.2 Study-design filter – You limited inclusion to RCTs after searching, but PRISMA diagram still counts 295 non-RCTs removed a priori. |
State the filter a priori in eligibility criteria; update PRISMA counts so numbers add up. |
2.3 Dual quality tools – PEDro and CASP overlap; RoB-2 is recommended for RCTs. |
Either justify the dual use (what extra insight did CASP add?) or use RoB-2 only. Provide an appendix with domain-level judgements. |
2.4 Data items – No code-book shows how you harmonised disparate adherence metrics (sessions vs minutes). |
Add a table describing how each raw metric was converted to the reported % adherence. |
2.5 Synthesis approach – Narrative synthesis chosen, yet you present pooled ranges in the abstract. |
State explicitly that meta-analysis was not feasible (give reasons: I² > x%, inconsistent metrics) and reference SWiM reporting guidance. |
3 Results
Issue |
Recommendation |
3.1 Study count / date mismatch – Text says trials 2014-2023; Table S1 lists 2021-2023 only; one ref is 2024. |
Re-audit the publication years; ensure Tables and text align. |
3.2 Adherence range interpretation – A single outlier (21 %) drives the lower bound; no median or IQR reported. |
Provide median ± IQR; consider a forest-style plot in Supplement to visualise variability. |
3.3 Drop-out data – Reasons are pooled qualitatively; denominator (per group) often absent. |
Report attrition as n/N (%) for each arm; classify reasons under CONSORT categories (withdrawal, AE, death, lost to follow-up). |
3.4 Tables S1–S3 formatting – Column headings wrap awkwardly; some cells contain multi-line paragraphs. |
Split complex variables (e.g., “GOLD stage & mean FEV₁”) into separate columns; lock row heights to improve readability. |
4 Discussion
Issue |
Recommendation |
4.1 Overlap with results – First three paragraphs restate numeric findings. |
Condense to 3–4 sentences; shift emphasis to interpretation (why heterogeneity persisted, how technology or programme intensity matters). |
4.2 Causality leap – Statement that “high satisfaction correlates with better adherence” is plausible but untested within included RCTs. |
Rephrase as hypothesis (“may contribute”) and cite external longitudinal evidence (e.g., Hoaas 2016 BMC Med Inform). |
4.3 Limitations – Good, but you omit publication bias and language bias (English/French/Spanish only). |
Add sentences covering these two biases and how they could influence the adherence range. |
4.4 Future perspective – Currently one paragraph; lacks specific research questions. |
Suggest concrete directions: consensus core-outcome set for adherence; automated device-captured adherence metrics; equity impact analyses. |
5 Conclusion
Issue |
Recommendation |
Over-generalises (“TPR was equally successful as PR”) despite admitted heterogeneity and moderate GRADE certainty. |
Temper wording: “Across eight small-to-moderate RCTs, TPR often achieved adherence and QoL outcomes comparable to PR, but inconsistent definitions preclude firm conclusions.” |
6 Language & style
- Several long, multi-clause sentences (e.g., Introduction lines 52-63) obscure meaning. Break into shorter, active-voice statements.
- Ensure consistent tense (Results in past tense; Discussion partly drifts to present).
- Replace abbreviations on first use in each major section (e.g., PR, TPR, QoL).
- Duplicate reference numbers (e.g., 16 appears twice also as ref 14). Re-number sequentially.
- DOIs missing for refs 32, 35, 37.
- Figure 1 (PRISMA): Resolution is low; axis text unreadable. Export at ≥300 dpi.
- GRADE Table S6: Cells “⊕⊕◯◯” appear mis-aligned with outcomes; add a legend.
- Several long, multi-clause sentences (e.g., Introduction lines 52-63) obscure meaning. Break into shorter, active-voice statements.
- Ensure consistent tense (Results in past tense; Discussion partly drifts to present).
- Replace abbreviations on first use in each major section (e.g., PR, TPR, QoL).
Author Response
Thank you for reviewing our thesis and your insightful and constructive comments. We greatly appreciate your feedback, which has been extremely helpful in improving the quality of our work.
We have carefully considered all your suggestions and have made every effort to incorporate them into the revised version of the thesis. Your input has been invaluable in refining our arguments and improving the clarity of our analysis.
- Introduction
Issue |
Recommendation |
1.1 Conceptual framing – Adherence is introduced but not defined; you later lament the absence of a universal definition. |
Add an operational definition up-front (e.g., WHO, ISPOR or ATS) and state why heterogeneity hampers comparisons. Answer: OK: We have revised the text by incorporating the WHO definition and providing an explanation of the reasons behind the heterogeneity. |
1.2 Redundancy / length – Multiple sentences repeat global COPD burden figures; section spans >650 words. |
Compress global-burden statistics to one sentence; dedicate space to conceptual gaps (e.g., why previous reviews did not resolve adherence heterogeneity). Answer: OK: We try to compress the global-burden. Previous reviews may not have fully addressed the heterogeneity in adherence; however, we believe that this issue extends beyond the scope of the current introduction. Our primary aim here is to provide an overview of the existing literature, concentrating on how adherence is measured and the extent to which patients follow treatment, rather than delving into the reasons behind the variety of measurement methods. This is an important point that should be elaborated on in more detail in future work.
|
1.3 Citations – Some references are duplicated (e.g., refs 16/41) or missing DOIs. |
De-duplicate and cross-check each citation; supply DOIs per journal style. Answer: OK: Citations checked and duplicate citations removed |
1.4 Rationale / objectives – Objective blends adherence, satisfaction, QoL but does not signal hierarchy. |
End with a clear, bullet-style aim: “Primary: quantify adherence; Secondary: compare drop-out and satisfaction; Tertiary: examine QoL.” Answer: OK; Completed |
|
|
2. Methods |
|
2.1 Search strategy transparency – The string is paraphrased and the Boolean logic is unclear. |
Provide the full PubMed strategy (search line-by-line, with date of final run) in Supplement S1; replicate for other databases. Answer: We endeavoured to clarify Boolean logic.
|
2.2 Study-design filter – You limited inclusion to RCTs after searching, but PRISMA diagram still counts 295 non-RCTs removed a priori. |
State the filter a priori in eligibility criteria; update PRISMA counts so numbers add up. Answer: OK: We have made modifications Thank you for your comment regarding the handling of RCTs in the PRISMA diagram. The confusion stemmed from a misstatement in our initial description. We did not apply an automatic filter to exclude RCTs during the database search. All articles were initially retained, and RCTs were excluded manually during the first screening step based on our inclusion/exclusion criteria. We have corrected the text and the PRISMA diagram to accurately reflect this process.
|
2.3 Dual quality tools – PEDro and CASP overlap; RoB-2 is recommended for RCTs. |
Either justify the dual use (what extra insight did CASP add?) or use RoB-2 only. Provide an appendix with domain-level judgements. Answer: OK: We explained why we use both PEDro and CASP. |
2.4 Data items – No code-book shows how you harmonised disparate adherence metrics (sessions vs minutes). |
Add a table describing how each raw metric was converted to the reported % adherence. Answer: We did not develop a codebook to harmonise adherence metrics across studies, as the original researchers did not provide consistent or complete raw data. The definition and measurement of adherence varied significantly between studies, and in many instances, key methodological details were absent. Consequently, it was not feasible to redefine or standardise adherence using a common framework. Therefore, we reported adherence percentages exactly as they were presented in the original publications. We were unable to provide a table detailing how each raw metric was converted into the reported percentage adherence, as these percentages had already been calculated and presented in the original studies. We reported them as they were in our thesis, and in most cases, the primary studies did not specify the exact method used to compute adherence. |
2.5 Synthesis approach – Narrative synthesis chosen, yet you present pooled ranges in the abstract. |
State explicitly that meta-analysis was not feasible (give reasons: I² > x%, inconsistent metrics) and reference SWiM reporting guidance. Answer: OK: Completed |
3. Results |
|
3.1 Study count / date mismatch – Text says trials 2014-2023; Table S1 lists 2021-2023 only; one ref is 2024. |
Re-audit the publication years; ensure Tables and text align. Answer: We would like to clarify that Table 1 indeed includes studies from 2014 to 2023, as stated in the text. The first study listed in the table is from 2014, not 2021. Furthermore, no reference from 2024 is included in Table 1. |
3.2 Adherence range interpretation – A single outlier (21 %) drives the lower bound; no median or IQR reported. |
Provide median ± IQR; consider a forest-style plot in Supplement to visualise variability. Answer: We appreciate the suggestion to calculate an average adherence rate with a confidence interval. However, we would like to emphasise that the data on adherence across the included studies are highly inconsistent regarding how adherence is defined and measured. Some studies report adherence as the number of sessions per week, others as the number of sessions per day, and others as participation per session. Moreover, the total number of sessions and the study designs vary substantially. |
3.3 Drop-out data – Reasons are pooled qualitatively; denominator (per group) often absent. |
Report attrition as n/N (%) for each arm; classify reasons under CONSORT categories (withdrawal, AE, death, lost to follow-up). Answer: Reasons for drop-out were presented in Table 3 “results” for each study. |
3.4 Tables S1–S3 formatting – Column headings wrap awkwardly; some cells contain multi-line paragraphs. |
Split complex variables (e.g., “GOLD stage & mean FEV₁”) into separate columns; lock row heights to improve readability. Answer: We would like to clarify that in Table 1, the data regarding COPD severity and the mean FEV1 (which directly corresponds to COPD severity) have already been presented in two separate columns. Therefore, we do not fully understand the suggestion to further "split complex variables," as these two aspects are already clearly separated and presented independently in the table. |
4 Discussion
Issue |
Recommendation |
4.1 Overlap with results – First three paragraphs restate numeric findings. |
Condense to 3–4 sentences; shift emphasis to interpretation (why heterogeneity persisted, how technology or programme intensity matters). This issue was addressed and corrected, thank for the comment. |
4.2 Causality leap – Statement that “high satisfaction correlates with better adherence” is plausible but untested within included RCTs. |
Rephrase as hypothesis (“may contribute”) and cite external longitudinal evidence (e.g., Hoaas 2016 BMC Med Inform). This issue was addressed and corrected, thank for the comment. |
4.3 Limitations – Good, but you omit publication bias and language bias (English/French/Spanish only). |
Add sentences covering these two biases and how they could influecnce the adherence range. This issue was addressed and corrected, thank for the comment. |
4.4 Future perspective – Currently one paragraph; lacks specific research questions. |
Suggest concrete directions: consensus core-outcome set for adherence; automated device-captured adherence metrics; equity impact analyses. This issue was addressed and corrected, thank for the comment. |
5 Conclusion
Issue |
Recommendation |
Over-generalises (“TPR was equally successful as PR”) despite admitted heterogeneity and moderate GRADE certainty. |
Temper wording: “Across eight small-to-moderate RCTs, TPR often achieved adherence and QoL outcomes comparable to PR, but inconsistent definitions preclude firm conclusions.” Done
We replaced abbreviations on first use in each major section as recommended.
DOI modified.
In Grade Table S6, we have already created a legend for this table: Polo et al. reported a number of participants with inconsistency b. All authors used different methods to quantify adherence, except for three (30,34,36), leading to inconsistency (consult Table 3). c. Quality of life is not reported using the same scales between studies (refer to Table 1). d. Fewer than 300 participants were involved. ⊕⊕⊕⊕ indicates high certainty of evidence, ⊕⊕⊕O represents moderate certainty of evidence, ⊕⊕OO signifies low certainty, and ⊕OOO reveals very low certainty. Concerned the symbols: They appear misaligned despite our best efforts to adjust them, and unfortunately, we have not been able to improve their alignment further. |
6 Language & style
All issues regarding language and style were corrected, thank you for the comment.
- Several long, multi-clause sentences (e.g., Introduction lines 52-63) obscure meaning. Break into shorter, active-voice statements.
- Ensure consistent tense (Results in past tense; Discussion partly drifts to present).
- Replace abbreviations on first use in each major section (e.g., PR, TPR, QoL).
- Duplicate reference numbers (e.g., 16 appears twice also as ref 14). Re-number sequentially.
- DOIs missing for refs 32, 35, 37.
- Figure 1 (PRISMA): Resolution is low; axis text unreadable. Export at ≥300 dpi.
- GRADE Table S6: Cells “⊕⊕◯◯” appear mis-aligned with outcomes; add a legend.
Reviewer 3 Report
Comments and Suggestions for Authors
To the authors:
The study is well-designed and properly conducted. The manuscript’s content is straightforward, and the conveyed message is important. I have some minor comments that will add to the manuscript.
Introduction
A general comment is that the authors provide a well-written introduction section with excellent background.
Methods
The authors provide an adequately described methods section and the SR protocol was registered in PROSPERO.
The PRISMA guidelines is provided adequately. However, I suggest the authors to remove from Figure 1. the first box of screening, “Records after duplicate removed (n=70)” since this information is provided at the next box (Records screened n=70).
At the paragraph of “Search strategy and study selection”, please rationalize why you decide to check databases for the last decade (since 2014). It was a coincidence or is it supported (?).
The “Data Extraction” section should be rewritten in a methodological way. Authors do not need to refer to Tables 1, 2, 3, since all tables should be presented in the section of the results. I advise you to describe THE WAY of data extraction. For example you can start in such way: “Data from the enrolled studies was extracted using study-specific forms with the following domains: authors, publication date, …, intervention characteristics, such as program session frequency…., as well primary secondary and tertiary outcomes in three Tables.
Similarly, at the paragraph “Quality Assessment” rather than using the words “Table S4” and “Table S5” here, describe that the quality assessment is summarized in Tables. Then in the section of the results you will name the Tables S4 and S5.
Similarly, at the paragraph “Certainty of Evidece”, remove the word “Table S6”.
Did authors follow a process when outcome data were not reported in an included study? For example, did you contact the corresponding authors to provide additional data? If yes, they included this process in the methods used, they should write it in this paragraph of the section.
Results
In general, the section is clear and well written
The author write: “Only RCTs and publications from the last ten years were selected, yielding 97 articles.” is the number 97 correct? Please check.
In the paragraph of “Results of tertiary outcomes”, authors write:
- Based on other QoL scales (mMRC, CAT, Euro-QoL-5), two studies (35,36, 36) found no significant difference in QoL improvements according to MCID for one of them (35). In the first parenthesis, there are three numbers; remove one. The last sentence is a bit confusing. Could you please clarify if no significant differences in QoL found in both studies 35 and 36?
- “All participants enrolled in the studies contributed to the conclusions, except for one (34)”. Could you please clarify?
The paragraph “Certainty of evidence” should be removed from the section of the Results. If authors want to include details for GRADE quality assessment, they can include it in the section of the METHODS. In the section of the Results, they can write only that the GRADE results are presented in Table S6.
Discussion
The discussion section is well-written and substantiated. I have no concerns, it flows very nicely.
A minor comment in the line of the first paragraph is to replace the words “ aims to analyse” with the word “analysed”
Author Response
Thank you for your valuable feedback and your time. We have endeavoured to incorporate and adapt these suggestions to enhance the quality and rigor of this thesis.
-
The study is well-designed and properly conducted. The manuscript’s content is straightforward, and the conveyed message is important. I have some minor comments that will add to the manuscript.
Thank you very much for reviewing our thesis and for your insightful and constructive comments. We greatly appreciate your feedback, which has been extremely helpful in improving the quality of our work.
We have carefully considered all your suggestions and have made every effort to incorporate them into the revised version of the thesis. Your input has been invaluable in refining our arguments and improving the clarity of our analysis.
The relevant modifications are highlighted in yellow in the manuscript.
Introduction
A general comment is that the authors provide a well-written introduction section with excellent background.
In the comments regarding “search strategy and study selection”, we explained why we decided to check databases for the last decade.
Methods
The authors provide an adequately described methods section and the SR protocol was registered in PROSPERO.
In the comments regarding “search strategy and study selection”, we explained why we decided to check databases for the last decade.
The PRISMA guidelines is provided adequately. However, I suggest the authors to remove from Figure 1. the first box of screening, “Records after duplicate removed (n=70)” since this information is provided at the next box (Records screened n=70).
Comments regarding the “Results” section: Yes, the figure of 97 articles identified during the search conducted in April is accurate. We have re-examined all our search strategy drafts and confirmed that the process indeed produced 97 articles.
At the paragraph of “Search strategy and study selection”, please rationalize why you decide to check databases for the last decade (since 2014). It was a coincidence or is it supported (?).
Thank you for the question. The objective was to gather data from the reent articles, and considering the novelty of the argument the availability of articles before that data was restricted.
The “Data Extraction” section should be rewritten in a methodological way. Authors do not need to refer to Tables 1, 2, 3, since all tables should be presented in the section of the results. I advise you to describe THE WAY of data extraction. For example you can start in such way: “Data from the enrolled studies was extracted using study-specific forms with the following domains: authors, publication date, …, intervention characteristics, such as program session frequency…., as well primary secondary and tertiary outcomes in three Tables.
Similarly, at the paragraph “Quality Assessment” rather than using the words “Table S4” and “Table S5” here, describe that the quality assessment is summarized in Tables. Then in the section of the results you will name the Tables S4 and S5.
Similarly, at the paragraph “Certainty of Evidece”, remove the word “Table S6”.
Thanks for your comment, we improved the method’s description.
Did authors follow a process when outcome data were not reported in an included study? For example, did you contact the corresponding authors to provide additional data? If yes, they included this process in the methods used, they should write it in this paragraph of the section.
Interesting question, even if the studies presented heterogenous outcomes we agreed that it was not needed contacting the articles authors.
Results
In general, the section is clear and well written
The author write: “Only RCTs and publications from the last ten years were selected, yielding 97 articles.” is the number 97 correct? Please check.
In the paragraph of “Results of tertiary outcomes”, the authors write:
- Based on other QoL scales (mMRC, CAT, Euro-QoL-5), two studies (35,36, 36) found no significant difference in QoL improvements according to MCID for one of them (35). In the first parenthesis, there are three numbers; remove one. The last sentence is a bit confusing. Could you please clarify if no significant differences in QoL found in both studies 35 and 36?
- “All participants enrolled in the studies contributed to the conclusions, except for one (34)”. Could you please clarify?
Thank you for your comments, the issues raised in the item 1 and 2 where clarified in the manuscript.
The paragraph “Certainty of evidence” should be removed from the section of the Results. If authors want to include details for GRADE quality assessment, they can include it in the section of the METHODS. In the section of the Results, they can write only that the GRADE results are presented in Table S6.
Comments regarding the “Certainty of evidence” section have been modified as recommended.
Discussion
The discussion section is well-written and substantiated. I have no concerns, it flows very nicely.
Thank you for the encouragement.
A minor comment in the line of the first paragraph is to replace the words “ aims to analyse” with the word “analysed”
We changed “aims to analyse” to “analysed” in the discussion section as recommended.
Round 2
Reviewer 2 Report
Comments and Suggestions for Authors
The revised manuscript addresses virtually all major issues raised in the first round. The authors have:
-
inserted a clear operational definition of adherence and re-framed the objectives into primary / secondary / tertiary hierarchy;
-
clarified the a-priori RCT filter and updated the PRISMA counts;
-
justified the dual PEDro + CASP appraisal;
-
supplied fuller search syntax and improved narrative synthesis transparency;
-
tightened language, removed most duplicate references and added missing DOIs;
-
upgraded tables (now in-text rather than Supplement only) and added a legend to GRADE.
Remaining minor issues below
# | Section | Comment & required action (minor) |
1 | Introduction | Paragraph duplication. The WHO definition of adherence and ensuing sentences appear twice back-to-back (lines “WHO defined adherence as…” to “…method of quantification”) – delete the duplicate block. |
2 | Keywords | Typo: “Telerehabilitaton” → “Telerehabilitation”. |
3 | Methods 2.6 / headings | Sub-heading numbering jumps 2.4 → 2.6 (no 2.5) – renumber sequentially. |
4 | Figures | PRISMA flow (Figure 1) still exports at ~150 dpi; axis text blurs when zoomed. Re-export ≥300 dpi. |
5 | Style | Several long multi-clause sentences remain (e.g., Results 3.5 first sentence spans >60 words). Consider one additional copy-edit pass for brevity & active voice. |
6 | References | Spot-check again for residual duplicates (e.g., refs 20/21 very similar) and ensure all include DOIs. |
The quality of English is fine
